**Data Availability Statement:** All relevant data are within the manuscript and its Supporting

# Population-level prevalence of detectable HIV viremia in people who inject drugs (PWID) in Ukraine: Implications for HIV treatment and case finding interventions

Yana Sazonova[1]*, Roksolana Kulchynska[2], Marianna Azarskova[2], Mariia Liulchuk[3], Tetiana Salyuk[4], Ivan Doan[2], Ezra Barzilay[2]

1 PEPFAR Coordination Office in Ukraine, Division of Global HIV and TB, U.S. Centers for Disease Control and Prevention, Kyiv, Ukraine, 2 Division of Global HIV and TB, U.S. Centers for Disease Control and Prevention, Kyiv, Ukraine, 3 State Institution "The L.V. Gromashevskij Institute of Epidemiology and Infectious Diseases of NAMS of Ukraine", Kyiv, Ukraine, 4 Monitoring and Evaluation Unit, ICF "Alliance for Public Health", Kyiv, Ukraine

* yana.sazonova2@gmail.com

## Abstract

Achievement of viral load suppression among people living with HIV is one of the most important goals for effective HIV epidemic response. In Ukraine, people who inject drugs (PWID) experience the largest HIV burden. At the same time, this group disproportionally missed out in HIV treatment services. We performed a secondary data analysis of the national-wide cross-sectional bio-behavioral surveillance survey among PWID to assess the population-level prevalence of detectable HIV viremia and identify key characteristics that explain the outcome. Overall, 11.4% of PWID or 52.6% of HIV-positive PWID had a viral load level that exceeded the 1,000 copies/mL threshold. In the group of HIV-positive PWID, the detectable viremia was attributed to younger age, monthly income greater than minimum wage, lower education level, and non-usage of antiretroviral therapy (ART) and opioid agonistic therapy. Compared with HIV-negative PWID, the HIV-positive group with detectable viremia was more likely to be female, represented the middle age group (35–49 years old), had low education and monthly income levels, used opioid drugs, practiced risky injection behavior, and had previous incarceration history. Implementing the HIV case identification and ART linkage interventions focused on the most vulnerable PWID sub-groups might help closing the gaps in ART service coverage and increasing the proportion of HIV-positive PWID with viral load suppression.

## Introduction

In times when antiretroviral therapy (ART) is available and proven to be an effective treatment for HIV infection, HIV viral load suppression (VLS) remains the most crucial indication of successful treatment programs [1]. The approach "Treatment as Prevention" (TasP) and the

Information files. The dataset is restricted to the number of variables and cases that were used for analysis

**Funding:** The bio-behavioral surveillance study has been supported by the President's Emergency Plan for AIDS Relief (PEPFAR) through the Centers for Disease Control and Prevention (CDC) under the terms of Cooperative Agreement NU2GGH000840 implemented by Alliance for Public Health, awarded to YS, TS, and ML. The U.S. Centers for Disease Control and Prevention (CDC) Ukraine country's office provided technical support for the study design, protocol, and instruments development, and conducted continued project oversight. The BBS protocol, instruments, and this paper were reviewed and approved by CDC Scientific Integrity Branch. The authors have not received any specific funding for the data analysis presented in this manuscript and its development. The findings and conclusions of this paper are those of the authors and do not necessarily represent the official position of the CDC

**Competing interests:** The authors have declared that no competing interests exist.

slogan "Undetectable = Untransmittable" (U = U) underscores the importance of suppressed viral load at individual and population levels in decreasing HIV morbidity, mortality, and transmission [2–5]. Globally, the estimated size of people living with HIV (PLHIV) reached 38.4 million in 2021 [6]. Of all PLHIV, 75% (66–85%) were accessing ART, and 68% (60–78%) were virally suppressed [6]. Key populations, especially people who inject drugs (PWID), have experienced disproportional access to ART and, as a result, a suboptimal level of viral load suppression [7].

In Ukraine, the HIV epidemic is primarily concentrated in key populations (KPs) and people who inject drugs (PWID), representing more than one-third of PLHIV enrolled in HIV care in 2020 [8]. According to the latest estimates, there were 350,300 PWID in Ukraine, and 22.6% had HIV [9, 10]. This group has substantial gaps in access to care at each stage of the HIV care continuum. The latest data show that 58% of PWID living with HIV know their status, 70% are on antiretroviral therapy, and 74% are virally suppressed [11]. These indicators are much lower than that of the general population in Ukraine, which includes 69% of PLHIV who are aware of their status, 83% of diagnosed PLHIV are on ART, and 94% of ART patients are virally suppressed, and fall far below UNAIDS 95-95-95 goals [8, 12]. Low ART coverage and poor adherence endanger the country's effective implementation of the TasP strategy and could contribute to increasing the amount of PLHIV with detectable viremia (DV), facilitating active HIV transmission, and increasing HIV incidence and death rates [13–15].

The bi-annual Bio-Behavioral Surveillance (BBS) studies are conducted in Ukraine in the PWID population to assess HIV prevalence, HIV incidence, and behavioral characteristics. The 2017 BBS study in PWID was unique as it collected the blood samples in all participants identified as HIV-positive to conduct subsequent HIV viral load (VL) laboratory testing. In this paper, it allowed us to investigate the population-level prevalence of detectable HIV viremia (PDV) in PWID on the city level and as an aggregated estimate for the major cities in Ukraine. In addition, the analysis aimed to measure correlates of the DV among PWID.

## Materials and methods

The following subsections of the paper provide a brief overview of the study setting, design, sample, data collection procedures, and ethical considerations. More detailed information on the survey methods, procedures, and data collection approaches can be found elsewhere [11].

### Study setting, design, and sample

We used the data from the BBS study among PWID, conducted in 30 cities in Ukraine from October 3rd to December 28th, 2017. The study collected data on behavioral practices with on-site testing for HIV and HCV infections and laboratory testing for HIV recency and VL. Survey cities were selected to represent all first-order administrative division units of Ukraine, which included Kyiv city (Ukraine's capital), capital cities of 24 Ukrainian oblasts (regions), six smaller cities that had epidemiological significance, and Sevastopol city (Autonomic Republic of Crimea).

The sample size for each city was calculated using previous knowledge of city-level HIV prevalence in PWID, design effect for Respondent Driven Sampling (RDS) studies, and standard error. As a result, cities' samples ranged from 200 to 550 participants. The total cities-pooled sample size was 10,076 PWID.

The study eligibility criteria included PWID who had recent (30 days) injection drug use practices, resided in the surveyed cities, and were ≥15 years old.

## Data collection

Data collection was achieved through three different mechanisms: questionnaire-based interview, biomarker testing for HIV, hepatitis C (HCV), and laboratory testing of dry blood spots (DBS) for HIV recency and VL.

An interviewer-assisted standardized questionnaire was used to collect self-reported data on socio-demographic characteristics, behavioral practices, and HIV service use. The questionnaire included the questions from the previous BBS rounds to calculate key indicators for UNAIDS Global AIDS Monitoring Country Report, HIV treatment cascade indicators and to measure other key variables which explain the HIV epidemic in PWID in Ukraine based on the national BBS working group recommendations [16–19].

All study participants were tested for HIV and HCV despite their previous knowledge about HIV or HCV statuses. Testing was performed according to the manufacturer's instructions and study standard operational procedures (SOPs). The HIV testing algorithm included the Profitest test HIV 1/2 (the first rapid test (RT)), SD Bioline HIV 1/2 3.0 (second HIV RT), and Alere Determine™ HIV-1/2 (third HIV RT). All HIV-negative test results of the first RT were reported to the participants. All participants with HIV-positive results of the first HIV RT were asked to provide a capillary blood sample (500 μL) using a K2-EDTA microtainer tube. This sample was used to perform confirmatory HIV RTs and fill in the dry blood spot (DBS) cards for further laboratory analysis for HIV recency and VL. Additionally, all participants were tested for HCV antibodies using the Profitest HCV RT. All participants received pre-and post-test counseling by medical workers according to the national guidelines.

Viral load testing of collected DBS was performed using the Abbott RealTime HIV-1 assay with the automated extraction and detection m2000 system platform (Abbott Molecular Inc., USA) at the laboratory of molecular virology of the State Institution "The L.V. Gromashevskij Institute of Epidemiology and Infectious Diseases of the National Academy of Medical Sciences (NAMS) of Ukraine" [20, 21].

The HIV and HCV rapid testing results and laboratory VL testing results were merged with the data from the survey questionnaire using the study participants' unique codes. In all study cities, the data collection teams established the referral pathways to the specialized healthcare facilities for participants with HIV and HCV rapid test positive results. After entering the healthcare facility, the study participants received the same opportunities for further diagnostic testing and treatment as any other patients. Additionally, all participants, including HIV-negative, received information about community-based HIV services for PWID available in the city and were referred there.

## Ethical considerations

The data collection teams obtained written informed consent from all participants before the study. The study eligibility criteria included participants = > 15 years old. The study procedures were developed to ensure additional ethical considerations for participants under the age of 18 years old. Parental consent was not required according to the study protocol with reference to the Ukrainian code of professional ethics [22]. Additionally, the data collection teams acted to ensure that signed informed consent was clearly understood and involved only volunteer-based participation. All participants were informed that they could withdraw from the study at any moment. Also, the study team was trained to provide active referrals to relevant services in case of identification of any violence, child trafficking, or sexual exploitation. The study used monetary incentives for participation in the study (the equivalent of 6 USD) and recruitment of peers (the equivalent of 1.5 USD per peer).

The study was approved by the Ethical Committee at the Ukrainian Institute of Public Health Policy, Kyiv, Ukraine. Also, this project was reviewed in accordance with U.S. Centers for Disease Control and Prevention (CDC) human research protection procedures and was determined to be a research, but CDC investigators did not interact with human subjects or had access to identifiable data or specimens for research purposes.

### Measurements

**Outcome.** The HIV viral load was selected as a key outcome for this analysis. The outcome was categorized into three groups based on HIV and VL test results: (1) HIV-positive with DV group ($\geq$1,000 copies/mL), (2) virally suppressed HIV-positive group (<1,000 copies/mL), and (3) HIV-negative group. The VL threshold for variable categorization was chosen based on the World Health Organization (WHO) classification [18]. In total, 2,261 DBS samples were collected from HIV-positive study participants, 2,214 DBS samples were tested to determine VL level, and 47 DBS samples were rejected by the VL laboratory's specialists due to quality issues.

**Covariables.** The covariables included socio-demographic characteristics, aggregated injection and sexual behavioral practices, and usage of HIV prevention and treatment services.

The socio-demographic information included variables that can show the differences in the profiles of PWID with DV compared to virally suppressed (VLS) and HIV-negative groups. In particular, we included sex (male and female), age categorized in years (<35, 35–49, and = >50), marital status (single and married or live with a sexual partner), monthly income (less than minimum wage, higher than minimum wage but less than country's average, and higher the country's average), and educational level (completed nine years of school or less, completed secondary education (11 years of school or college), bachelor and graduate degrees or higher degree).

The behavioral variables present aggregated injection and sexual high-risk practices. The high-risk injection practice included questions about sharing the syringe or needle in the past 30 days, sharing other injection instruments for drug cooking or distribution in the past 30 days, and purchasing the drug in the prefilled syringe in the past 30 days. The variable was categorized with "yes" if the participant reported at least one of those practices. Similarly, the high-risk sexual practices variable was aggregated data from responses to several questions, i.e., having sex with an HIV-positive partner(s) in the past 90 days, having more than five sexual partners in the past 90 days, and having a commercial sexual partner(s) in the past 90 days. High-risk sexual behavior was classified as "yes" if the participant reported at least one of those sexual practices.

We defined injection behavior using variables representing the type of injection drugs used in the past 30 days. We grouped all the injection drug types into three categories. "Only opioids" included heroin, desomorphine, homemade opioids, and synthetic opioids. "Only stimulants" had amphetamines, methamphetamines, cocaine, and "salt". "Mixed or other" group included other or multiple drug types related to several categories.

Previous incarceration experience was computed using two questions "Have you ever been incarcerated?" and "If yes, when was the last time you were released?". Based on answers to those two questions, the variable was grouped into three categories: "have never been incarcerated", "released from prison less than a year ago", and "released from prison more than a year ago".

HIV services utilization PDV's correlates encompassed self-reported data about the usage of the HIV-prevention and treatment services. In particular, HIV prevention services were presented by two variables: (1) experience of receiving harm reduction services in the past 12 months ("yes", "no"), and (2) experience of receiving the opioid agonistic therapy (OAT)

currently or ever in the lifetime ("yes", "no"). HIV treatment variables were used to explain the PDV only in the HIV-positive subsample. Those covariables included receiving community-based PLHIV case management services ("yes", "no"), and aggregated variables of ARV's drugs use and time on ART ("on ART <6 months", "on ART ≥6 months", "on ART (time is NA)", "Not on ART"). Additionally, we included the awareness of HIV-positive status ("aware", "unaware", "missing/not disclosed"), which was computed based on the responses of HIV-positive participants to the cascade of questions: "Have you ever been tested for HIV?", "Have you received your HIV test result?", "Can you tell us the most recent HIV test result?", "If yes, what was it?". However, this variable was excluded from the multivariable analysis to prevent collinearity with the ART use variable.

### Statistical analysis

The population-level PDV estimates were calculated for each of the study cities in R (version 4.1.3, 2022-03-10) using the "RDS" package [23]. The PDV estimates for each city were adjusted using Gile's SS weighting coefficients. The aggregated PDV estimate was calculated on the pooled sample for 30 cities with adjustments to the RDS weights and the city-level population size estimates.

For the descriptive and bivariable analysis, we looked at characteristics of PWID stratified by the study outcome categories: HIV-positive group with DV, HIV-positive group who achieved VLS, and HIV-negative group. The chi-square test for categorical variables and the t-test for a continuous variable (age) were used to measure the association between the outcome and key covariables.

We ran two separate models to produce unadjusted and adjusted prevalence ratios (PR) to explain the variation in study outcome using defined covariables. We chose Poisson regression with robust variance to receive direct PR and reduce the bias of artificially overestimated odds ratios [24, 25]. We used the parsimonious models with the backward selection approach and removed all covariables not associated with the outcome at the significance level for p<0.15. The models were assessed using the chi-square goodness of fit test and models' assumptions. The models were performed in R, version 4.0.3, package "sandwich" [26].

We did not use the RDS-weights in the bivariable and multivariable analyses due to a lack of consensus for the regression models that combine multiple cities [27]. However, the RDS diagnostic results of the city samples are available elsewhere [11].

### Results

In the study, 7,815 (77.2%) participants received HIV-negative results, and 2,261 (22.8%) received HIV-positive testing results. Subsequent laboratory testing of HIV-positive samples confirmed that 11.4% of the total PWID population, or 52.6% of HIV-positive PWID, had a detectable level of viral load (Fig 1). Population-level of PDV varied substantially across the Ukrainian cities showing the range from 23.6% (95% CI: 18.8%-28.4%) in Kryvyi Rih (Dnipropetrovska oblast) to 0% in Uzhgorod (Zakarpatska oblast) (Fig 2).

Table 1 investigates the distribution of the key characteristics of the HIV-positive PWID with DV compared to the HIV-positive PWID with achieved VLS and HIV-negative PWID. The HIV-positive PWID with achieved VLS was the oldest on average compared to the group with DV and HIV-negative group (mean age 40.1 years old vs. 37.6 years old and 34.6 years old, respectively). Generally, the PWID population in each defined group was predominantly male. However, the HIV-negative group included larger proportions of male representatives than the HIV-positive group (83.8% vs. 73.5% in the virally suppressed group and 75.3% in the group with DV). The HIV-positive group with suppressed viral load had the largest proportion

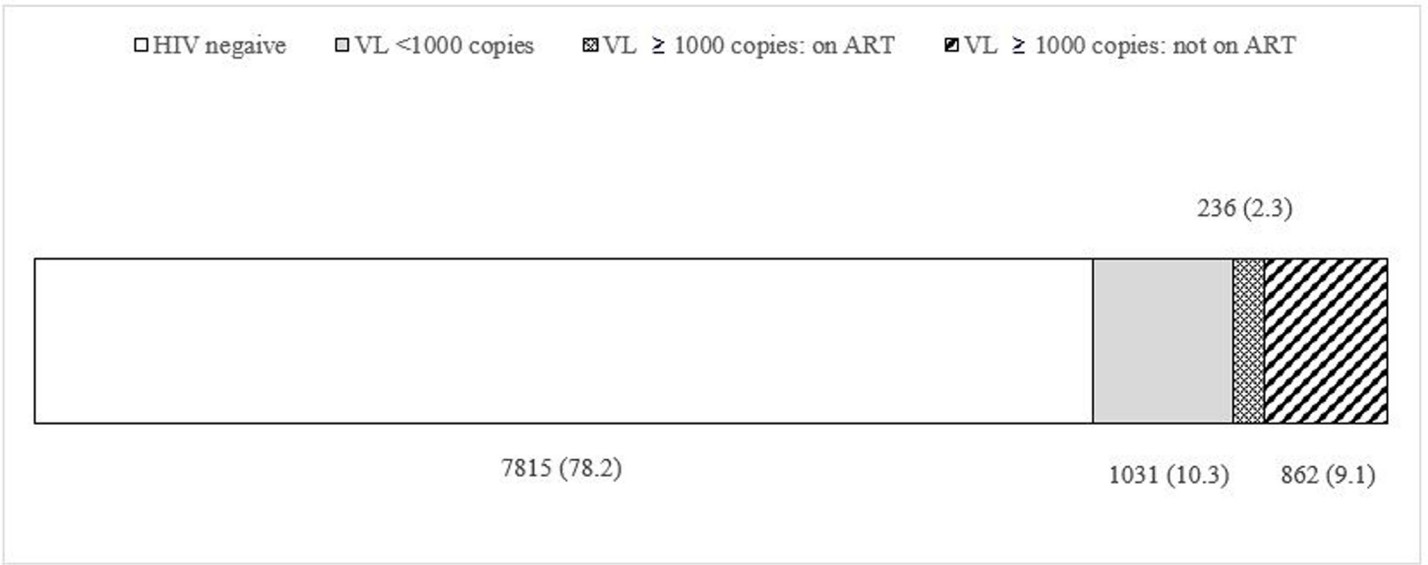

**Fig 1. HIV VL results in distribution by self-reported ART status, n\* (%‡).** \* Picture 1 does not include participants who received HIV-positive results but did not provide their ART status information during the interview or whose DBS samples were rejected by the laboratory for VL testing due to the samples' quality issues (NA = 132). ‡ Percentages are presented as population estimates that were adjusted for the RDS study design.

of PWID with a monthly income below minimum wage compared to the two other groups. The HIV-positive group with DV had the lowest representation of PWID with a bachelor or graduate education level (12.1% vs. 15.0% in the suppressed group and 21.1% in the HIV-negative group). In addition, the HIV-positive group with DV had the largest proportion of those who were recently released from incarcerated facilities (7.9% vs. 6.7% in the suppressed group and 4.9% in the HIV-negative group).

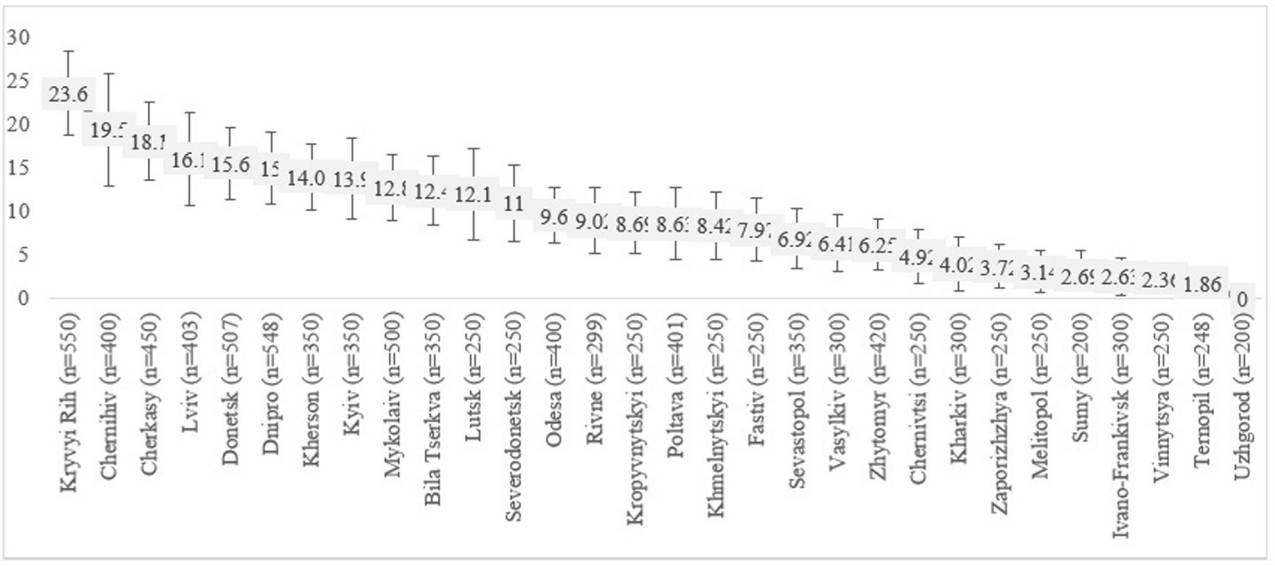

**Fig 2. City-level variations in PDV among all PWID, %.**

**Table 1. Descriptive and bivariate statistics for participants characteristics by VL groups.**

| Characteristic | HIV positive with VL <1000 copies/mL‡ | HIV positive with VL = >1000 copies/mL‡ | p-value* | HIV negative‡ | p-value** |
|---|---|---|---|---|---|
| *HIV prevention and treatment services* | | | | | |
| Subgroup sample size | 1,031 | 1,183 | | 7,815 | |
| Receive harm reduction services (%) | | | | | |
| yes | 680 (64.9) | 617 (51.4) | <0.001 | 3,583 (45.1) | <0.001 |
| no | 351 (35.1) | 566 (48.6) | | 4,232 (54.9) | |
| Receive community-based PLHIV case management services (%) | | | | | |
| yes | 141 (12.7) | 111 (9.7) | 0.030 | - | - |
| no | 890 (87.3) | 1,072 (90.3) | | - | - |
| Receive OAT (%) | | | | | |
| yes | 255 (23.4) | 138 (10.8) | <0.001 | 637 (7.4) | <0.001 |
| no | 776 (76.6) | 1,045 (89.2) | | 7,178 (92.6) | |
| Awareness of HIV-positive status | | | | | |
| aware | 708 (67.3) | 549 (45.2) | <0.001 | - | - |
| unaware | 276 (28.4) | 549 (48.4) | | - | - |
| *missing/not disclosed* | *47 (4.3)* | *85 (6.3)* | | - | - |
| Receive ART (%) | | | | | |
| on ART <6 months | 70 (7.0) | 44 (3.6) | <0.001 | - | - |
| on ART ≥6 months | 504 (47.2) | 159 (12.9) | | - | - |
| on ART (time is NA) | 85 (8.1) | 33 (2.4) | | - | - |
| Not on ART | 325 (33.4) | 862 (74.8) | | - | - |
| *missing information* | *47 (4.3)* | *85 (6.3)* | | - | - |
| *Behavioral practices* | | | | | |
| High-risk sexual behavior (%) | | | | | |
| yes | 380 (37.2) | 522 (44.6) | 0.001 | 4,309 (56.3) | <0.001 |
| no | 651 (62.8) | 661 (55.4) | | 3,506 (43.7) | |
| High-risk Injection behavior (%) | | | | | |
| yes | 474 (47.2) | 578 (51.8) | 0.030 | 3,000 (41.0) | <0.001 |
| no | 557 (52.8) | 605 (48.2) | | 4,815 (59.0) | |
| Type of injection drug in the past month (%) | | | | | |
| mixed-use or other | 223 (21.5) | 284 (24.0) | 0.314 | 1,974 (25.0) | <0.001 |
| only opioids | 753 (72.7) | 818 (69.8) | | 4,788 (61.0) | |
| only stimulants | 55 (5.8) | 81 (6.3) | | 1,053 (14.0) | |
| *Socio-demographic characteristics* | | | | | |
| Age in years (mean (SD)) | 40.07 (6.63) | 37.56 (6.78) | <0.001 | 34.62 (7.96) | <0.001 |
| Age categories: <35 years old | 206 (20.1) | 397 (33.2) | <0.001 | 4170 (53.3) | <0.001 |
| Age categories: 35–49 years old | 729 (71.2) | 734 (62.2) | | 3263 (41.9) | |
| Age categories: ≥50 years old | 96 (8.6) | 52 (4.6) | | 382 (4.8) | |
| Sex (%) | | | | | |
| male | 754 (73.5) | 898 (75.3) | 0.342 | 6,588 (83.8) | <0.001 |
| female | 277 (26.5) | 285 (24.7) | | 1,227 (16.2) | |
| Marital status (%) | | | | | |
| Single | 473 (46.3) | 546 (46.2) | 0.977 | 3,279 (41.6) | 0.024 |
| Married or living together with a sexual partner | 558 (53.7) | 637 (53.8) | | 4,536 (58.4) | |
| Monthly income (%)*** | | | | | |

*(Continued)*

**Table 1.** (Continued)

| Characteristic | HIV positive with VL <1000 copies/mL‡ | HIV positive with VL = >1000 copies/mL‡ | p-value* | HIV negative‡ | p-value** |
|---|---|---|---|---|---|
| below minimum wage | 362 (32.8) | 279 (21.7) | | 1,384 (16.8) | |
| higher than the minimum wage but less than the country's average | 574 (57.2) | 752 (63.3) | <0.001 | 5,408 (68.3) | <0.001 |
| higher than the country's average | 95 (10.0) | 152 (15.0) | | 1,021 (14.9) | |
| missing | 0 (0.0) | 0 (0.0) | | 2 (0.0) | |
| Educational level (%) | | | | | |
| school or less | 180 (17.2) | 252 (21.9) | | 1,264 (16.1) | |
| complete secondary education | 685 (67.8) | 774 (66.0) | | 4,893 (62.7) | |
| bachelor | 82 (6.9) | 95 (7.0) | 0.002 | 950 (12.1) | <0.001 |
| graduate | 83 (8.1) | 62 (5.1) | | 703 (9.0) | |
| missing information | 1 (0.1) | 0 (0.0) | | 5 (0.1) | |
| *Other* | | | | | |
| Experience of incarceration (%) | | | | | |
| have never been incarcerated | 415 (39.6) | 526 (44.2) | | 4,938 (63.2) | |
| released from prison less than a year ago | 60 (6.7) | 85 (7.9) | 0.023 | 385 (4.9) | <0.001 |
| released from prison more than a year ago | 556 (53.7) | 570 (47.8) | | 2490 (31.9) | |
| missing information | 0 (0.0) | 2 (0.1) | | 2 (0.0) | |

‡ Percentages are presented as population estimates that were adjusted for the RDS study design

\* p-value for comparison of HIV-positive groups of PWID for VL<1000 copies and VL ≥1000 copies (chi-square test for all categorical variables and t-test for continuous, e.g., "age in years".

\*\* p-value for VL ≥1000 copies and HIV negative (chi-square test for all categorical variables and t-test for continuous, e.g., "age in years".

"-" variable was not used for the HIV-negative subgroup of PWID.

\*\*\* Minimum wage in Ukraine as of December 2017 was 3,200 UAH (approximately 116 USD using 2017's currency exchange rate); Average salary in Ukraine as of December 2017 was 8,777 UAH (approximately 320 USD using 2017's currency exchange rate).

The high-risk injection behavior was significantly higher in the HIV-positive PWID group with DV (51.8%) than in the suppressed group (47.2%) and in the HIV-negative group (41.0%). Almost half (44.6%) of HIV-positive PWID with DV had reported at least one type of high-risk sexual practice in the past three months. This indicator was less prevalent among PWID with suppressed VL (37.2%). However, the HIV-negative PWID had an even higher prevalence of high-risk sexual practices (56.3%).

The use of harm reduction services and OAT were more prevalent in the HIV-positive group of PWID with suppressed VL. The same tendency was observed for the uptake of HIV-related case management services and ART in HIV-positive PWID subgroups. The HIV-positive status awareness was significantly higher in the group of HIV-positive PWID with achieved VLS than PWID with DV (67.3% vs. 45.2%, p<0.001). Additionally, a larger proportion of PWID with DV decided not to disclose their HIV status during the interview compared to PWID with VLS (6.3% vs. 4.3%).

The multivariable analysis (Table 2) in the HIV-positive group confirmed that PWID with DV, in comparison to the group with VLS, were younger, more likely to have middle-level monthly income (higher than minimum wage but less than the country average), and had lower (school or less) education levels compared to graduate education level. Having DV also was explained by non-usage of OAT and ART services. PWID with no experience of participation in the OAT program were 12% more likely to have DV than PWID who had such experience. Being on ART substantially increases chances for VLS, even PWID with ARVs usage

**Table 2. Crude and adjusted prevalence ratios for key characteristics associated with DV.**

| Covariables | HIV-positive PWID with unsuppressed VL vs. suppressed VL | | HIV-positive PWID with unsuppressed VL vs. HIV-negative | |
|---|---|---|---|---|
| | Crude PR* | Adjusted PR, 95% CI | Crude PR | Adjusted PR, 95% CI |
| Receive harm reduction services: yes [ref.: no] | **0.77 [0.71–0.83]** | NS | **1.25 [1.12–1.38]** | NS |
| Receive community-based PLHIV case management services: yes [ref.: no] | **0.81 [0.70–0.93]** | NS | - | - |
| Receive OAT: yes [ref.: no] | **0.61 (0.53–0.70)** | **0.88 (0.77–0.99)** | **1.40 (1.19–1.65)** | NS |
| Awareness of HIV-positive status: aware [ref.: unaware] | **0.66 (0.61–0.71)** | ** | - | - |
| Awareness of HIV-positive status: missing information [ref.: unaware] | 0.97 (0.84–1.11) | ** | - | - |
| Receive ART: on ART <6 months [ref.: not on ART] | **0.53 (0.42–0.67)** | **0.55 (0.43–0.69)** | - | - |
| Receive ART: on ART ≥ 6 [ref.: not on ART] | **0.33 (0.29–0.38)** | **0.36 (0.31–0.41)** | - | - |
| Receive ART: on ART (time is NA) [ref.: not on ART] | **0.39 (0.29–0.52)** | **0.41 (0.31–0.55)** | - | - |
| Receive ART: missing information [ref.: not on ART] | 0.89 (0.78–1.01) | 0.89 (0.78–1.01) | - | - |
| High-risk sexual behavior: yes [ref.: no] | **1.15 (1.06–1.24)** | *NS* | **0.68 (0.61–0.76)** | **0.67 (0.60–0.75)** |
| High-risk injection risks of HIV transmission [ref.: no] | 1.06 (0.98–1.14) | *NS* | **1.45 (1.30–1.61)** | **1.42 (1.28–1.57)** |
| Age: 35–49 years old [ref.: <35 years old] | **0.76 (0.71–0.82)** | **0.85 (0.79–0.91)** | **2.11 (1.88–2.37)** | **1.75 (1.56–1.97)** |
| Age: ≥50 years old [ref.: <35 years old] | **0.53 (0.43–0.67)** | **0.64 (0.52–0.79)** | **1.38 (1.05–1.81)** | 0.89 (0.67–1.19) |
| Male sex [ref.: female] | 1.07 (0.98–1.18) | *NS* | **0.64 (0.56–0.72)** | **0.53 (0.47–0.60)** |
| Marital status: Single [ref.: married or live with the partner] | 1.01 (0.93–1.09) | *NS* | **1.16 (1.04–1.29)** | *NS* |
| Monthly income: higher than minimum wage but less than country average [ref.: below minimum wage] | **1.30 (1.18–1.44)** | **1.12 (1.02–1.22)** | **0.73 (0.64–0.83)** | **0.85 (0.75–0.97)** |
| Monthly income: higher than country average [ref.: below minimum wage] | **1.41 (1.24–1.61)** | **1.12 (0.99–1.26)** | **0.77 (0.64–0.93)** | 1.01 (0.84–1.21) |
| Educational level: complete school or vocational school [ref.: school or less] | **0.91 (0.83–0.99)** | 0.96 (0.88–1.04) | **0.82 (0.72–0.94)** | 0.92 (0.80–1.04) |
| Educational level: bachelor [ref.: school or less] | 0.92 (0.79–1.08) | 0.99 (0.86–1.13) | **0.55 (0.44–0.68)** | **0.67 (0.54–0.84)** |
| Educational level: graduate [ref.: school or less] | **0.73 (0.60–0.90)** | **0.80 (0.67–0.96)** | **0.49 (0.37–0.63)** | **0.62 (0.48–0.81)** |
| Type of injection drug in the past month: only opioids use [ref.: only stimulators use] | 0.87 (0.76–1.01) | *NS* | **2.04 (1.64–2.54)** | **1.60 (1.15–1.84)** |
| Type of injection drug in the past month: mixed-use or other [ref.: only stimulants use] | 0.94 (0.80–1.10) | *NS* | **1.76 (1.39–2.23)** | **1.45 (1.29–1.99)** |
| Experience of incarceration: Released from prison ≤ 1 year ago [ref.: have never been incarcerated] | 1.05 (0.90–1.21) | *NS* | **1.88 (1.52–2.32)** | **1.78 (1.45–2.20)** |
| Experience of incarceration: Released from prison >1 year ago [ref.: have never been incarcerated] | **0.91 (0.84–0.98)** | *NS* | **1.93 (1.73–2.16)** | **1.72 (1.54–1.94)** |

*PR–prevalence ratio "-" variable was not used for the HIV-negative subgroup of PWID.

NS–Non-significant covariable / was excluded from the final model.

** Covariable "Awareness of HIV-positive status" was excluded from multivariable analysis due to collinearity with the "Receive ART" covariable.

experience of fewer than six months were 45% less likely to have DV than HIV-positive PWID who were not on ART. We have not found statistically significant associations between high-risk injection and sexual behavioral practices with VL level and excluded those covariables from the final parsimonious model.

The second multivariable model (Table 2) presented the crude and adjusted PRs for HIV-positive PWID with DV compared to the HIV-negative group. In the adjusted analysis, HIV-positive PWID with DV were older, less likely to be male, had middle-level monthly income (higher than minimum wage but less than country average) compared to below minimum wage group, and had bachelor or graduate education degree compared to "school or less" education level group. HIV-positive PWID with DV were more likely to use opioid or mixed injection drugs compared to exclusive stimulants use. Also, they were more likely to practice high-risk injection behavior in the past month and have some incarceration experience. The usage of OAT and harm reduction services has not proved to have a significant association in the multivariable model and was excluded at some point in the multivariable analysis. The ART use and HIV case-management covariables were excluded from the model due to their inclusiveness only to HIV-positive patients.

## Discussion

Effective HIV/AIDS response requires the achievement of nearly universal ART coverage and sustaining the viral suppression in the PLHIV population [28, 29]. Having a large proportion of the PLHIV population with an unsuppressed viral load poses substantial risks of active HIV transmission that might result in increasing HIV incidence [30]. Ukraine has eliminated any CD4 thresholds for ART initiation and committed to implementing the Test and Start Strategy [31, 32]. Currently, there are no legal barriers to providing ART to all HIV-diagnosed patients immediately after a confirmed HIV diagnosis. However, PWID in Ukraine are disproportionally underserved within the medical system and face the additional barriers of stigma and discrimination because of injection drug use and HIV-positive status [10, 33].

To the best of our knowledge, this is the first study to measure the population-level prevalence of detectable viremia in PWID in Ukraine and identify key characteristics that might explain variation. Our study follows the recently published analysis of the HIV treatment cascade among PWID [11]. However, our study focuses on PDV in the total PWID population to show gaps in controlling the HIV epidemic in the PWID population and preventing HIV transmission. Such findings provide valuable information about the PWID profile, key behavioral characteristics, and coverage with HIV prevention, case finding, and treatment services.

Our analysis revealed that more than 11.4% of all PWID and more than half (53%) of all HIV-positive PWID in Ukraine had DV ($\geq$ 1,000 copies/ml). Among them, nearly 55% were unaware of their HIV-positive status and had never initiated ART. Although such findings demonstrate the large gap in achieving the UNAIDS target of 95% viral suppression among PLHIV [12], it is far above that of most other countries with published VL coverage results among PWID [34]. Additionally, we found that the population viral load is not geographically heterogeneous, indicating the "hot spot" areas for the high risk of probable HIV transmission in the PWID population within Ukraine.

Numerous studies explained the elevated risk of DV in PLHIV by factors such as insufficient time of ART use, poor ART adherence, late HIV diagnosis, single marital status, ART treatment modifications, lower BMI baseline status, and history of immunological failure [35, 36]. Due to the secondary nature of the data, our analysis was restricted to the available variables in the dataset. However, we confirmed the association between DV and several identified factors in similar studies. Firstly, we reiterated that the absence of ART treatment is a key

predictor for DV, and PWID with greater time on ART had greater chances to achieve VLS. However, even PWID, who reported being on ARVs for less than six months, had significantly greater chances to achieve VLS than PWID who were not on ART. Such findings align with general knowledge about the ART effect when the median time to achieve viral load suppression is estimated to be 181 days after treatment initiation [37]. Additionally, this time can be even shorter when more efficient ARV regimens are available. For instance, ART patients who use INSTI-based regimens may achieve VLS with a median time of sixty days after treatment initiation [38].

Additionally, we underscored the importance of the OAT treatment for PWID to achieve viral load suppression. Another study from Ukraine has already investigated the OAT effect towards improvements in the HIV care continuum in PWID living with HIV [39]. That study confirmed the positive association of OAT with HIV status awareness and ART initiation in the PWID population. However, it failed to receive conclusive results about the OAT effect on optimal ART adherence, possibly due to insufficient sample size. Nevertheless, our finding aligns with numerous other studies that indicate that OAT improves ART adherence and, as a result, VLS in PWID [40–44].

We found the disproportionate burden of DV in several PWID subgroups. The results suggest that among the HIV-positive group, younger PWID, those with average or high monthly income and a low education level were less likely to be virally suppressed. Nearly 73% of HIV-positive PWID with DV generally did not engage in ART. Of this group, 81%-85% on average is characterized by young persons (<35 years old), those with low education, and those with reported higher than national average monthly income. The ART linkage program might develop more focused approaches to engage representatives of those groups on ART. Conversely, 13.4% of HIV-positive PWID with DV reported ART use for over six months. Medical workers and community-based case managers might decide to provide additional ART adherence packages of services to facilitate the achievement of VLS in that group.

Almost half (46.4%) of HIV-positive PWID with DV were unaware of their HIV-positive status. Such a finding might be considered a clear message for the HIV case-finding efforts. We believe that our analysis of HIV-positive PWID with DV compared to the HIV-negative group might help differentiate the HIV case-finding approaches and bring HIV testing services to communities with larger needs. The identified differences in the socio-demographic profile of HIV-positive PWID with DV compared to the HIV-negative group might inform HIV case-finding programs and close gaps in the knowledge of HIV-positive status. HIV case-finding programs might consider using more sensitized approaches to involve female PWID, those who have low educational and income levels, and use opioid injection drugs.

The study confirmed that PWID with DV were more likely to experience incarceration. This is in line with the other studies, which found that all HIV care continuum outcomes improve during incarceration; however, there is a decline after the release [45]. Additional efforts might be taken to link HIV-positive PWID after their release from incarceration facilities to ART sites in the civil sector to ensure efficient linkage and continuation of treatment.

We found that HIV-positive PWID with DV were more likely to practice high-risk injection behavior than HIV-negative PWID. At the same time, we cannot confirm the significant association between harm reduction programs and PDV. The harm reduction services were widely available in Ukraine at the time of the BBS study. According to the program data, the services were provided to 226,467 PWID in 2017, which resulted in 64% of the estimated PWID population coverage [46]. The expansion of such services with a focus on PWID with DV might facilitate the reduction of PWID who practice high-risk injection behavior and break the chain of HIV transmission. Additionally, we found that PWID with DV were less likely to practice high-risk sexual behavior than the HIV-negative group. Such findings might

inform the Pre-Exposure Prophylaxis (PrEP) program and speed up its expansion to eligible HIV-negative PWID.

## Limitations

This study is not without limitations. Due to the cross-sectional nature of the survey data, we cannot confer the causality effect between the identified factors and HIV viral load levels [47]. The self-reported behavioral data might raise a concern of social desirability bias due to under-reporting the high-risk behavior in a face-to-face interview [48]. The self-reported data may also contain other types of informational biases. In particular, some participants might fail to identify themselves as clients of harm reduction or OAT programs if they have not recently received such services. The analysis was conducted using secondary data, and we were limited to available variables. For instance, some other baseline clinical indications (e.g., CD4, comorbidities, body mass index (BMI), mental health), events of ART interruptions, or some structural barriers (working schedule, stigma, and discrimination in the clinical setting) might explain additional variation in the study outcome.

The study sampling strategy might introduce several limitations as well. The study was conducted in the large cities of Ukraine, which may not effectively represent the entire PWID population in the country. However, the study included 30 different cities representing all of Ukraine's regions. Also, the HIV epidemic in Ukraine is highly urbanized, as nearly 70% of HIV cases were found in urban areas, a trend that has been stable for decades [8]. Additionally, a large sampling size and RDS approach might reduce the selection bias and increase the study's external validity [49].

## Conclusions

The study found that every tenth PWID in Ukraine has detectable HIV viremia, which presents substantial challenges for implementing the TasP program strategy and achieving HIV epidemic control. The study identified key predictors of DV that might inform HIV prevention, case finding, and treatment service and help to sensitize program approaches to access HIV-positive PWID subgroups that experience a larger risk of DV.

## Supporting information

**S1 Dataset.**
(CSV)

## Acknowledgments

We thank the study participants for sharing the information about their experience. We appreciate the active participation and leadership of the Public Health Center of the Ministry of Health Ukraine in the implementation of the survey and dissemination of findings. We thank Alliance Consultancy and data collection teams for their work throughout the study.

**Disclaimer:** The findings and conclusions in this manuscript are those of the authors and do not necessarily represent the official position of the funding agencies.

## Author Contributions

**Conceptualization:** Yana Sazonova, Roksolana Kulchynska, Marianna Azarskova, Mariia Liulchuk, Tetiana Salyuk, Ivan Doan, Ezra Barzilay.

**Data curation:** Yana Sazonova, Marianna Azarskova, Mariia Liulchuk.

**Formal analysis:** Yana Sazonova.

**Investigation:** Tetiana Salyuk.

**Methodology:** Yana Sazonova, Roksolana Kulchynska, Marianna Azarskova, Mariia Liulchuk, Tetiana Salyuk, Ivan Doan, Ezra Barzilay.

**Writing – original draft:** Yana Sazonova, Roksolana Kulchynska.

**Writing – review & editing:** Roksolana Kulchynska, Marianna Azarskova, Mariia Liulchuk, Tetiana Salyuk, Ivan Doan, Ezra Barzilay.

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
