## [Decision Letter · Decision Letter 0]

4 May 2023

PONE-D-23-05654Population-level prevalence of detectable HIV viremia in people who inject drugs in Ukraine: implications for HIV treatment and case finding interventions.PLOS ONE

Dear Dr. Sazonova,

Thank you for submitting your manuscript to PLOS ONE. After careful consideration, we feel that it has merit but does not fully meet PLOS ONE’s publication criteria as it currently stands. Therefore, we invite you to submit a revised version of the manuscript that addresses the points raised during the review process. A well written manuscript. Please attend to all comments presented below by both reviewers. Additionally, try to review references that are "online articles" by adding DOI or link to ease access. Please review PLOS ONE reference guidelines for details Submission Guidelines | PLOS ONE and ensure the revised submission fully comply with PLOS ONE publication criteria.

We look forward to receiving your revised manuscript.

Kind regards,

Ibrahim Abdullahi Jahun, PhD., MSc., MD.

Academic Editor

PLOS ONE

Journal Requirements:

“The bio-behavioral surveillance study has been supported by the President's Emergency Plan for AIDS Relief (PEPFAR) through the Centers for Disease Control and Prevention (CDC) under the terms of Cooperative Agreement NU2GGH000840 implemented by Alliance for Public Health.”

Reviewers' comments:

Reviewer's Responses to Questions

**Comments to the Author**

1. Is the manuscript technically sound, and do the data support the conclusions?

Reviewer #1: Yes

Reviewer #2: Yes

2. Has the statistical analysis been performed appropriately and rigorously? 

Reviewer #1: I Don't Know

Reviewer #2: Yes

3. Have the authors made all data underlying the findings in their manuscript fully available?

Reviewer #1: Yes

Reviewer #2: Yes

4. Is the manuscript presented in an intelligible fashion and written in standard English?

Reviewer #1: Yes

Reviewer #2: Yes

5. Review Comments to the Author

Reviewer #1: General

This is a well written paper. It presents on an important issue in achieving the 90-90-90 – viral load suppression. By presenting prevalence of detectable viraemia and its correlates, the information could help drive efforts at achieving the 3rd 95 of the UN strategy among HIV-positive PWIDs in Ukraine.

The comments I provide here are to help the reader of the paper when published to better understand the information provided

Abstract

Line 33/34. In describing the relationship between HIV-negative PWID and HIV-positive PWID, consider using the phrase “compared with” in place of the word “unlike”.

Materials and methods

• General: This section has all the information needed, but the arrangement can be confusing. It appears the methods described here are reported in a previous study by Sazanova et al (2020; https://journals.plos.org/plosone/article?id=10.1371/journal.pone.0244572#pone-0244572-t002). If this is the case, authors are advised to refer the reader of this manuscript to the paper by Sazanova for details of the methods and endeavor to summarize it here, following the chronology and clarity in the referenced paper.

• However, I provide further specific comments below:

Study setting and design

• The chronological presentation of the paragraphs can be improved for better logical flow.

• The section includes sample size determination and calculation, which is not covered in the sub-heading. Consider a new section on sample size or expand the present sub-heading to include it – e.g “Study setting, design and sample”.

• Lines 71/73 – Part of the sentence here is a repetition of the first sentence in this section. Consider reviewing this sentence to remove redundancy.

• Lines 79/80 – Unless the aim here is to demonstrate the role of incentives in the recruitment, I would suggest that this issue is presented under the “ethical considerations” section.

Data collection

• General – one can deduce that what is presented here is a set of procedures used to collect data for the study. In chronology as presented, this includes blood collection and biomarker testing, and questionnaire-based interview. For better clarity, the authors may wish to consider presenting these separate paragraphs. Issues about how the data from the separate parts of the study was linked and referral services provided for HIV and HCV-positive participants can be discussed subsequently. But it is also important to mention what happened to HIV and HCV-negative participants. A sentence on pre and posttest counselling can be added in chronological flow with referral services to make the appropriate point.

• Line 82/83 – In the light of the general observation above, the authors may wish to revise the opening sentence to be more specific about the data collection algorithm alone. For example – “Data collection was achieved through three different mechanisms – questionnaire-based interview, blood collection, and biomarker testing for HIV, HCV and VL.” A chronology that presents the process followed in the field to a greater extent would provide more clarity.

• Line 83 – did the authors mean to say, “obtaining informed consent”? It feels like informed consent written alone does not say what was done.

Ethical considerations

Line 121 – I think the authors are missing the word “be” in the following phrase: “procedures and was determined to research”

Results

Line 215 – In presenting results on wage earning, it is important to be clear when mention is made of low or high wage. The tables (1 and 2) are clear of the income/wage classifications. Thus, if by “low monthly income” on line 215, the authors meant to say, “monthly income below minimum wage”, then using the right descriptor would help the reader more. This issue is observed in other parts of the paper for the authors’ consideration.

Lines 261/262 – It is not clear, the message that these phrases sought to convey – “had average monthly income in reference to lower income level” and “had any educational level than completed school”. The authors should kindly review and provide better clarity as necessary.

Discussion

Line 286 – While it is accurate that more than 11.1% of all PWIDs were reported to have DV in this paper, when compared to the actual reported estimate of 11.4%, this statement could lead the audience to wrong conclusions. Authors should consider revising this statement to avoid such risks.

Lines 316-319 – The two sentences here can be improved upon (Nearly 73% of HIV-positive PWID with DV generally did not engage in ART. This proportion among the young age group (<35 years old), subgroup with low education level, and those who reported about monthly income higher than on average ranged from 81%-85%.). Consider instead the following:

“Nearly 73% of HIV-positive PWID with DV generally did not engage in ART. Of this group, 81%-85% on average is characterized by young persons (<35 years old), those with low education, and those with reported higher than national average monthly income”

This is how I am reading the two sentences. The authors should consider the most appropriate revision though.

Lines 369/370 – This phrase in the last sentence can be revised for better clarity “treatment service to sensitize their approaches to access populations with greater needs.”

Reviewer #2: The manuscript was well structured and just has some minimal adjustments required to make it better.

1. Line 46 – Add a sentence on the total number of PLHIV who are on ART.

2. Line 52 – Give an estimate of the total number of PWIDs in Ukraine before you write the proportion of these PWIDs that have HIV.

3. Line 70 – State the selection criteria and sampling methods used to select these 30 cities.

4. Data collection – State the protocol used for the data collection for PWIDs who already knew their HIV status and were already on ART. Did you repeat HIV rapid tests for them?

5. Results – For clients with detectable viral loads, compare if duration of ART treatment has a significant relationship with the level of viremia.

6. PLOS authors have the option to publish the peer review history of their article (what does this mean?). If published, this will include your full peer review and any attached files.

Reviewer #1: No

Reviewer #2: **Yes: **ODUME BETHRAND (DR)

---

## [Author Response · Author response to Decision Letter 0]

14 Jul 2023

Dear editors and reviewers, 

We are grateful for the constructive feedback provided by the reviewers, which has allowed us to improve the manuscript significantly. In response to the reviewers’ and editors’ comments, we made the following changes to our submission: 

Editors’ comments (EC):

EC: Please ensure that your manuscript meets PLOS ONE's style requirements, including those for file naming. The PLOS ONE style templates can be found at

Authors Response (AR): Thank you! We checked and updated the manuscript accordingly.

EC: Thank you for stating the following financial disclosure:

“The bio-behavioral surveillance study has been supported by the President's Emergency Plan for AIDS Relief (PEPFAR) through the Centers for Disease Control and Prevention (CDC) under the terms of Cooperative Agreement NU2GGH000840 implemented by Alliance for Public Health.”

AR: Financial disclosure was updated to include option “d” and was added to the cover letter. The financial disclosure is following: 

“The BBS has been supported by the President's Emergency Plan for AIDS Relief (PEPFAR) through the Centers for Disease Control and Prevention (CDC) under the terms of Cooperative Agreement NU2GGH000840 implemented by the Alliance for Public Health. The authors received no specific funding for this work.”

Additionally, per your comment, we have deleted the financial disclosure from the manuscript text.

We note that you have indicated that data from this study are available upon request. PLOS only allows data to be available upon request if there are legal or ethical restrictions on sharing data publicly. For more information on unacceptable data access restrictions, please see http://journals.plos.org/plosone/s/data-availability#loc-unacceptable-data-access-restrictions

Thank you for the guidance. We included the dataset as the supplemental material. Please be informed that the dataset was restricted to the number of variables and cases that was used in analysis. 

EC: We note that you have indicated that data from this study are available upon request. PLOS only allows data to be available upon request if there are legal or ethical restrictions on sharing data publicly. For more information on unacceptable data access restrictions, please see http://journals.plos.org/plosone/s/data-availability#loc-unacceptable-data-access-restrictions

AR: Thank you for the guidance. We included the dataset as the supplemental material. Please be informed that the dataset was restricted to the number of variables and cases that was used in analysis.

Reviewers’ comments: 

Reviewer 1 (R1): 

R1: This is a well written paper. It presents on an important issue in achieving the 90-90-90 – viral load suppression. By presenting prevalence of detectable viraemia and its correlates, the information could help drive efforts at achieving the 3rd 95 of the UN strategy among HIV-positive PWIDs in Ukraine. 

The comments I provide here are to help the reader of the paper when published to better understand the information provided.

AR: Thank you so much for your comments and suggestions. We tried to address all of them in the updated version of the manuscript.

R1:

Abstract

Line 33/34. In describing the relationship between HIV-negative PWID and HIV-positive PWID, consider using the phrase “compared with” in place of the word “unlike”.

AR: Thank you for suggestion, it is corrected (Line 34 in track-changed version)

R1:

Materials and methods

General: This section has all the information needed, but the arrangement can be confusing. It appears the methods described here are reported in a previous study by Sazanova et al (2020; https://journals.plos.org/plosone/article?id=10.1371/journal.pone.0244572#pone-0244572-t002). If this is the case, authors are advised to refer to the paper by Sazanova for details of the methods and endeavor to summarize it here, following the chronology and clarity in the referenced paper.

AR: The chronology of presented methods was updated according to reviewer’s suggestion and reference on previous paper with more detailed description of methods was provided (Line 74, Lines 91-137 in track-changed version)

R1: 

Study setting and design

The chronological presentation of the paragraphs can be improved for better logical flow. 

AR: The chronological presentation of the paragraphs was improved (Lines 91-137 in track-changed version)

R1: The section includes sample size determination and calculation, which is not covered in the sub-heading. Consider a new section on sample size or expand the present sub-heading to include it – e.g “Study setting, design and sample”.

AR: The sub-heading was changed to “Study setting, design and sample” (Line 75 in track-changed version)

R1: Lines 71/73 – Part of the sentence here is a repetition of the first sentence in this section. Consider reviewing this sentence to remove redundancy. 

AR: The sentence was updated “We used the data from the BBS study among PWID, conducted in 30 cities in Ukraine from October 3rd to December 28th, 2017. The study collected data on behavioral practices with on-site testing for HIV and HCV infections and laboratory testing for HIV recency and VL.” Lines 76-79 in trach-changed version.

R1: Lines 79/80 – Unless the aim here is to demonstrate the role of incentives in the recruitment, I would suggest that this issue is presented under the “ethical considerations” section.

AR: It was moved to Ethical Considerations sub-section, lines 147-149 in track-changed version.

R1: 

Data collection

General – one can deduce that what is presented here is a set of procedures used to collect data for the study. In chronology as presented, this includes blood collection and biomarker testing, and questionnaire-based interview. For better clarity, the authors may wish to consider presenting these separate paragraphs. 

AR: The chronology was changed to represent study algorithm: questionnaire, rapid testing, lab testing. Each of them is presented as separate paragraphs Lines 90-122 in track-changed version.

R1: Issues about how the data from the separate parts of the study was linked and referral services provided for HIV and HCV-positive participants can be discussed subsequently. 

AR: The information was added “The HIV and HCV rapid testing results and laboratory VL testing results were merged with the data from the survey questionnaire using the study participants' unique codes.” Lines 123-124 in track-changed version.

R1: But it is also important to mention what happened to HIV and HCV-negative participants. 

AR: The information was added “Additionally, all participants, including HIV-negative, received information about community-based HIV services for PWID available in the city and were referred there.” Lines 128-130 in track-changed version.

R1: A sentence on pre and posttest counselling can be added in chronological flow with referral services to make the appropriate point.

AR: Information was added “All participants received pre-and post-test counseling by medical workers according to the national guidelines.” Lines 112-113 in track changed version.

R1: Line 82/83 – In the light of the general observation above, the authors may wish to revise the opening sentence to be more specific about the data collection algorithm alone. For example – “Data collection was achieved through three different mechanisms – questionnaire-based interview, blood collection, and biomarker testing for HIV, HCV and VL.” A chronology that presents the process followed in the field to a greater extent would provide more clarity.

AR: Opening sentence was added “Data collection was achieved through three different mechanisms: questionnaire-based interview, biomarker testing for HIV, hepatitis C (HCV), and laboratory testing of dry blood spots (DBS) for HIV recency and VL.” Lines 92-96 in track changed version. 

R1: Line 83 – did the authors mean to say, “obtaining informed consent”? It feels like informed consent written alone does not say what was done.

AR: The sentence was modified to get more clarity “The data collection teams obtained written informed consent from all participants before the study.” Lines 139-140 in track changed version.

R1: 

Ethical considerations

Line 121 – I think the authors are missing the word “be” in the following phrase: “procedures and was determined to research”

AR: Yes, it was a typo – corrected. Line 152-153 in track changed version.

R1: 

Results

Line 215 – In presenting results on wage earning, it is important to be clear when mention is made of low or high wage. The tables (1 and 2) are clear of the income/wage classifications. Thus, if by “low monthly income” on line 215, the authors meant to say, “monthly income below minimum wage”, then using the right descriptor would help the reader more. This issue is observed in other parts of the paper for the authors’ consideration.

AR: Categories were changed to have more clarity throughout the paper. Lines 247-248, 249, 277-279, 295-298 in track changed version.

R1: Lines 261/262 – It is not clear, the message that these phrases sought to convey – “had average monthly income in reference to lower income level” and “had any educational level than completed school”. The authors should kindly review and provide better clarity as necessary.

AR: Categories were changed to have more clarity throughout the paper. Lines 247-248, 249, 277-279, 295-298 in track changed version.

R1: 

Discussion

Line 286 – While it is accurate that more than 11.1% of all PWIDs were reported to have DV in this paper, when compared to the actual reported estimate of 11.4%, this statement could lead the audience to wrong conclusions. Authors should consider revising this statement to avoid such risks.

AR: 11.1% was a typo. The correct result is 11.4% which is calculated on weighted data. Corrected. 

R1: Lines 316-319 – The two sentences here can be improved upon (Nearly 73% of HIV-positive PWID with DV generally did not engage in ART. This proportion among the young age group (<35 years old), subgroup with low education level, and those who reported about monthly income higher than on average ranged from 81%-85%.). Consider instead the following: “Nearly 73% of HIV-positive PWID with DV generally did not engage in ART. Of this group, 81%-85% on average is characterized by young persons (<35 years old), those with low education, and those with reported higher than national average monthly income”.

AR: Thank you for suggestion. The text was changed to “Nearly 73% of HIV-positive PWID with DV generally did not engage in ART. Of this group, 81%-85% on average is characterized by young persons (<35 years old), those with low education, and those with reported higher than national average monthly income.”. Lines 353-356 in track changed version. 

Thank you for suggestion. The text was updated accordingly Lines 353-356 in track changed version.

R1: Lines 369/370 – This phrase in the last sentence can be revised for better clarity “treatment service to sensitize their approaches to access populations with greater needs.” 

AR: Text was updated “The study identified key predictors of DV that might inform HIV prevention, case finding, and treatment service and help to sensitize program approaches to access HIV-positive PWID subgroups that experience a larger risk of DV”. Lines 409-411.

Reviewer #2 (R2)

R2: Line 46 – Add a sentence on the total number of PLHIV who are on ART

AR: The information was added in percentages. “Globally, the estimated size of people living with HIV (PLHIV) reached 38.4 million in 2021. Of all PLHIV, 75% (66-85%) were accessing ART, and 68% (60-78%) were virally suppressed.” Lines 47-49 in track changes version.

R2: Line 52 – Give an estimate of the total number of PWIDs in Ukraine before you write the proportion of these PWIDs that have HIV.

AR: The estimate of the PWID size was added: “According to the latest estimates, there were 350,300 PWID in Ukraine, and 22.6% had HIV.” Lines 54-55 in track changes version.

R2: Data collection – State the protocol used for the data collection for PWIDs who already knew their HIV status and were already on ART. Did you repeat HIV rapid tests for them?

AR: Yes, the full study algorithm was used for all participants despite their previous knowledge pf HIV or ART. Data collection team has no access to verify self-reported information.

R2: Results – For clients with detectable viral loads, compare if duration of ART treatment has a significant relationship with the level of viremia.

AR: Thank you, we checked this relationship using categorical variable of time on ART (<6 months and => 6 months). Only categorical variable was available for analysis.

Thank you for your work in revising our manuscript. We remain available to respond to any further questions or requests for information.

---

## [Decision Letter · Decision Letter 1]

13 Aug 2023

Population-level prevalence of detectable HIV viremia in people who inject drugs (PWID) in Ukraine: implications for HIV treatment and case finding interventions

PONE-D-23-05654R1

Dear Dr. Sazonova,

We’re pleased to inform you that your manuscript has been judged scientifically suitable for publication and will be formally accepted for publication once it meets all outstanding technical requirements.

Kind regards,

Ibrahim Jahun, MD, MSC, PhD

Academic Editor

PLOS ONE

Additional Editor Comments (optional):

Reviewers' comments:

Reviewer's Responses to Questions

**Comments to the Author**

1. If the authors have adequately addressed your comments raised in a previous round of review and you feel that this manuscript is now acceptable for publication, you may indicate that here to bypass the “Comments to the Author” section, enter your conflict of interest statement in the “Confidential to Editor” section, and submit your "Accept" recommendation.

Reviewer #3: All comments have been addressed

Reviewer #4: All comments have been addressed

2. Is the manuscript technically sound, and do the data support the conclusions?

Reviewer #3: Yes

Reviewer #4: Yes

3. Has the statistical analysis been performed appropriately and rigorously? 

Reviewer #3: Yes

Reviewer #4: I Don't Know

4. Have the authors made all data underlying the findings in their manuscript fully available?

Reviewer #3: Yes

Reviewer #4: Yes

5. Is the manuscript presented in an intelligible fashion and written in standard English?

Reviewer #3: Yes

Reviewer #4: Yes

6. Review Comments to the Author

Reviewer #3: The manuscript is technically sound and the authors have addressed all the comments raised by reviewer 2 in the earlier submission. The manuscript is now fit for acceptance and publication.

Reviewer #4: (No Response)

7. PLOS authors have the option to publish the peer review history of their article (what does this mean?). If published, this will include your full peer review and any attached files.

Reviewer #3: No

Reviewer #4: No

---

## [Editor Report · Acceptance letter]

17 Oct 2023

PONE-D-23-05654R1 

Population-level prevalence of detectable HIV viremia in people who inject drugs (PWID) in Ukraine: implications for HIV treatment and case finding interventions. 

Dear Dr. Sazonova:

I'm pleased to inform you that your manuscript has been deemed suitable for publication in PLOS ONE. Congratulations! Your manuscript is now with our production department. 

Kind regards, 

on behalf of

Dr. Ibrahim Jahun 

Academic Editor

PLOS ONE